



# **Enhanced internal tidal mixing in the Philippine Sea**
# **mesoscale environment**
Jia You[1, 3, 4], Zhenhua Xu[*1, 2, 3, 4], Qun Li[5], Robin Robertson[6], Peiwen Zhang[1, 3, 4],
Baoshu Yin[1, 2, 3, 4]
[1]CAS Key Laboratory of Ocean Circulation and Waves, Institute of Oceanology, Chinese Academy of
Sciences, Qingdao, China
[2]Pilot National Laboratory for Marine Science and Technology, Qingdao, China
[3]Center for Ocean Mega-Science, Chinese Academy of Sciences, Qingdao, China
[4]College of Earth and Planetary Sciences, University of Chinese Academy of Sciences, Beijing, China
[5]Polar Research Institute of China, Shanghai, China
[6]China-Asean College of Marine Science, Xiamen University Malaysia, Sepang, Malaysia
*Corresponding author*: Zhenhua Xu, Institute of Oceanology, Chinese Academy of Sciences; Email:
xuzhenhua@qdio.ac.cn.
**Abstract.**
Turbulent mixing in the ocean interior is mainly contributed by internal wave breaking; however, the
mixing properties and the modulation effects of mesoscale environmental factors are not well-known.
Here, the spatially inhomogeneous and seasonally variable diapycnal diffusivities in the upper
Philippine Sea were estimated from ARGO float data using a strain-based finescale parameterization.
Based on a coordinated analysis of multi-source data, we found that the driving processes for diapycnal
diffusivities mainly included the near-inertial waves and internal tides. Mesoscale features were
important in intensifying the mixing and modulating its spatial pattern. One interesting finding was that,
besides near-inertial waves, internal tides also contributed significant diapycnal mixing for the upper
Philippine Sea. The seasonal cycles of diapycnal diffusivities and their contributors differed zonally. In
the mid-latitudes, wind-mixing dominated and was strongest in winter and weakest in summer. In
contrast, tidal-mixing was more predominant in the lower-latitudes and had no apparent seasonal
variability. Furthermore, we provide evidence that the mesoscale environment in the Philippine Sea
played a significant role in regulating the intensity and shaping the spatial inhomogeneity of the
internal tidal mixing. The magnitudes of internal tidal mixing was greatly elevated in regions of
energetic mesoscale processes. The anticyclonic mesoscale features were found to enhance diapycnal
mixing more significantly than did cyclonic ones.
**Keywords**: Mixing, Internal tides, Mesoscale, the Philippine Sea



## 1. Introduction

Turbulent mixing can alter both the horizontal and vertical distributions of temperature and salinity
gradients. These then modulate the ocean circulation variability, both globally and regionally. Many
studies have shown the existence of a complicated spatiotemporal pattern of diapycnal mixing in the
ocean interior. Such mixing inhomogeneity can influence the hydrological characteristics, ocean
circulation variability and climate change. The breaking of internal waves is believed to be the main
contributor to the ocean's diapycnal mixing (eg. Liu et al., 2013, Robertson R., 2001). Thus, clear
understanding the spatial patterns and dissipation processes of broad-band internal waves is necessary
to clarify and depict the global ocean mixing climatology.
The long-wavelength internal waves in the ocean are mainly in the form of near-inertial internal
waves (NIWs) and internal tides (eg. Alford and Gregg, 2001; Cao et al., 2018; Klymak et al., 2006),
and the internal solitary waves evolved from them also can trigger mixing (eg. Deepwell et al., 2017;
Grimshaw, et al., 2010; Shen et al., 2020). The wind-input NIW energy to the mixing layer is about
0.3-1.4 TW (eg. Alford, 2003; Liu et al., 2017; Rimac et al., 2013; Watanabe and Hibiya, 2002). The
NIW energy propagate downward, mainly dissipate and drive energetic mixing within the upper ocean
(Wunsch and Ferrari, 2004). Barotropic tidal currents flowing over rough topographic features can
generate internal tides (eg. Robertson R., 2001), with the global energy of 1.0 TW (Egbert and Ray,
2001; Jayne and St. Laurent, 2001; Song and Chen, 2020). Near the sources, the internal tidal mixing
intensify above the bathymetries, meanwhile, in the remote area, the tidal mixing is distributed
throughout the water column due to the multiple reflection and refraction processes. Therefore, the
relative contributions to the upper-layer diapycnal diffusivities by NIWs and internal tides should differ
regionally, which deserves further investigation.
In mid-latitudes, NIWs dominated the upper ocean mixing, as a result of the presence of westerlies
and frequent storms (eg. Alford et al., 2016; Jing et al., 2011; Whalen et al., 2018). However, from the
global view, the upper ocean mixing geography is inconsistent with the global wind field distribution.
For example, in low-latitudes, upper ocean mixing hotspots are located nearer to rough topographic
features, regardless of the wind conditions. This indicates that upper ocean mixing might be attributed
to non-wind-driven internal waves, such as internal tides. In order to better understand the ocean
mixing patterns and modulation mechanisms, we need to clarify the relative contributions between the



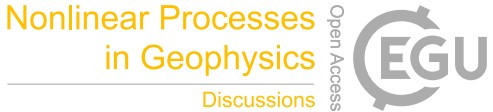

wind and tidal energy.
Internal tides are generally considered to be important to ocean mixing in the deep ocean, beyond the
influence of winds (Ferrari and Wunsch, 2009; Munk et al., 1998; MacKinnon et al., 2017). Many
factors influence the spatial pattern and energy transfer of internal tides. Higher-mode internal tides
break more easily near their sources, while the low-mode internal tides propagate long distances, even
thousands of kilometers. Propagating internal tides will be limited by several factors, such as
topography, stratification and turning latitude (eg. Vlasenko et al., 2013; Song and Chen, 2020;
Hazewinkel & Winters, 2011). Wave-wave interaction in the ocean also influences the spatiotemporal
variability of internal tides. For example, PSI (parametric subharmonic instability) is a potential avenue
to transfer internal tidal energy to other frequencies (Ansong et al., 2018). Moreover, stratification and
background flows also contribute to internal tidal spatial and temporal variability (eg. Karry et al., 2016;
Huang et al., 2018; Chang et al., 2019). Due to the complicated multi-scales of the background flows, it
is still unclear about how the background flow modulates the internal tides, their energy dissipation and
ocean mixing.
Recent research suggests that the mesoscale environment is a key factor influencing ocean mixing.
There is evidence that mesoscale eddies can enhance wind-driven mixing and internal tidal dissipation.
This enhancement will be more significant in the presence of an anticyclonic eddy (eg. Jing et al., 2011;
Whalen et al., 2018). Likewise, regional studies indicate that mesoscale features modulate the
generation and propagation of internal tides. Mesoscale currents can also broaden the range undergoing
internal tide critical latitude effects and enhance the energy transfer from diurnal frequencies to
semidiurnal or high frequencies (Dong et al., 2019). Mesoscale eddies are found to modulate internal
tide propagation (Rainville and Pinkel, 2006; Park and Watts, 2006; Zhao et al., 2010) and enable the
internal tide to lose its coherence (Nash et al., 2012; Kerry et al., 2016; Ponte and Klein, 2015).
Numerical simulation results support these observations (Kerry et al. 2014), indicating that the patterns
of internal tides is largely modulated by the position of eddies. An idealized numerical experiment
shows that the energy of internal tides shows bundled beams after passing through an eddy (Dunphy
and Lamb, 2014). And the mode-1 internal tides interactions with eddies will trigger higher-mode
signals. Up to now, research about mesoscale–internal tide interactions has been primarily focused on
the propagation pattern or 3-D structure of internal tides and has ignored their energy dissipation and





mixing effects. The latter is more important for altering the ocean circulation variability and climate
change.
The Philippine Sea, located in the Northwestern Pacific Ocean, is one of the most energetic internal
tidal regimes in the world. In this region, powerful internal tides significantly enhance ocean mixing, as
shown by numerical simulations (Wang et al., 2018). The importance of sub-inertial shear to ocean
mixing has been hypothesized from observations (Zhang et al., 2019), and the importance of internal
tides to mixing is supported through parameterization techniques (Qiu et al., 2012). On the other hand,
the Philippine Sea is an area with frequent typhoons, which make significant contributions to ocean
mixing. Multiple factors and mechanisms impact the turbulent mixing distribution in the Philippine Sea
(Wang et al., 2018). To date, it is unclear what the dominant factors are and how these factors modulate
the ocean mixing properties. Moreover, the role of mesoscale environment in regulating ocean mixing
is still not well understood.
At present, coupled numerical models are basically able to accurately simulate the generation and
propagation of internal tides. The internal tide dissipation and induced mixing are found to be
important for the determination of correct mixing parameterizations in numerical models (Robertson
and Dong, 2019). Some existing studies focus on the simulations of internal tidal breaking and tidally
induced mixing (Kerry et al., 2013; Kerry et al., 2014; Muller, 2013; Wang et al., 2018). It is difficult to
provide a complete spatial and temporal picture from direct observations of turbulence. This is due to
the scarcity of observations and their patchy distribution in time and space. Multisource data covering
multiple tidal cycles or preferably a spring-neap cycle, as well as a broad domain, are necessary to
acquire the spatiotemporal distribution and few of these have been collected. The development and
application of parameterization methods provide greater possibility of characterizing a broad-regional
mixing distribution and variability. A global pattern of ocean mixing has been provided using these
parameterization methods (Whalen et al. 2012; Kunze 2017). Furthermore, sensitivity studies have
been performed investigating the dependence of several factors to global mixing, such as bottom
roughness, internal tides, wind and background flows (eg. Whalen et al. 2012; Waterhouse. 2014;
Kunze and Eric. 2017; Whalen et al. 2018; Zhang et al. 2019). At present, parameterization is the most
effective method to investigate the modulation of tidal mixing by mesoscale background flows.
The spatial pattern and temporal variability of diapycnal diffusivities in the Philippine Sea are
examined in this paper. We provide evidence to verify the importance of tidal mixing in the upper layer





of this region. Moreover, we illustrate the modulation of mesoscale environment in tidal mixing
properties and distributions. Our data and methods are detailed in Section 2. Results and analysis,
including the spatial patterns and seasonal cycle of mixing, contributions of influencing factors and
internal tide-mesoscale interrelationships, are shown in Section 3. Finally the summary and discussion
are given in Section 4.

### 2.    Method and Data

**2.1    ARGO and Fine-scale parameterization method**

The ARGO Program is a joint international effort involving more than 30 countries and
organizations and having deployed over 15,000 freely drifting floats since 2000. The accumulated total
collected profiles exceeds 2 million profiles of conductivity, temperature, depth (CTD) along with other
geobiochemical parameters. The ARGO program has become the main data source for many research
and operational predictions of oceanography and atmospheric science (http://www.ARGO.ucsd.edu).
We screened the profiles from the Philippines Sea with quality control and estimated diapycnal
diffusivity and dissipation rate from them using a finescale parameterization.
The diapycnal diffusivity and turbulent kinetic energy dissipation rate can be estimated from a
fine-scale strain structure. This is based on a hypothesis that the energy can be transported from large to
small scales. In such scales, waves break due to shear or convective instabilities by weakly nonlinear
interactions between internal waves (Kunze et al., 2006). Presently, this method has been widely used
for the global ocean (eg. Wu et al., 2011; Kunze et al., 2017; Whalen et al., 2012; Fer et al., 2010;
Waterhouse et al., 2014). The dissipation rate $\varepsilon$ can be expressed as

$$\varepsilon = \varepsilon_0 \frac{\overline{N^2}}{N_0^2} \frac{\langle \xi_z^2 \rangle^2}{\langle \xi_{z\,GM}^2 \rangle^2} h(R_\omega) L(f, \overline{N}) \tag{1}$$

where $\varepsilon_0 = 6.73 \times 10^{-10} W/kg$ and $N_0 = 5.24 \times 10^{-3}/s$ , and $\overline{N^2}$ represents the averaged
buoyancy frequency of the segment. $\langle \xi_{z\,GM}^2 \rangle$ and $\langle \xi_z^2 \rangle$ are strain variance from the Garrett-Munk (GM)
spectrum (Gregg and Kunze, 1991) and the observed strain variance, respectively. The angle brackets
indicate integration over a specified range of vertical internal wavenumbers (see equations 4 and 5).
The function $h(R_\omega)$ accounts for the frequency content of the internal wave field and $R_\omega$ represents
shear/strain variance ratio. $R_\omega$ is fixed at 7, which is a global mean value (Kunze et al., 2006).

$$h(R_\omega) = \frac{1}{6\sqrt{2}} \frac{R_\omega(R_\omega+1)}{\sqrt{R_\omega-1}} \tag{2}$$


The function $L(f,\overline{N})$ corrects for a latitudinal dependence, here $f$ is the local Coriolis frequency,
and $f_{30}$ is the Coriolis frequency at $30°$, and $\overline{N}$ is the vertically averaged buoyancy frequency of the
segment.

$$L(f,\overline{N}) = \frac{f\,arcosh(\frac{\overline{N}}{f})}{f_{30}\,acrcosh(\frac{\overline{N}}{f_{30}})} \qquad (3)$$

strain $\xi_z$ was calculated from each segment,

$$\xi_z = \frac{N^2 - N_{ref}^2}{N^2} \qquad (4)$$

$$\langle \xi_z^2 \rangle = \int_{k_{min}}^{k_{max}} S_{str}(k_z)dk_z \leq 0.2 \qquad (5)$$

We derived $N$ from 2 to 10 dbar-processed temperature, salinity, and pressure data according to the
ARGO float resolution. $N_{ref}$, as a smooth piece-wise quadratic fit to the observed N profile, is fitted to
24 m. Here we remove segments that vary in the range of $\langle N^2 \rangle > 5 \times 10^{-4} s^{-2}$ or $\langle N^2 \rangle < 1 \times$
$10^{-9} s^{-2}$ since the strain signal at these levels is dominated by noise (Whalen et al., 2018). By
applying a fast Fourier transform (FFT) on half-overlapping 256 m segments along each vertical $\xi_z$
profile, we computed the spectra $S_{str}(k_z)$ and integrated them to determine the strain variance. We
integrated these spectra between the vertical wavenumbers $k_{min} = 0.003\,cmp$ and $k_{max} =$
$0.02\,cmp$ according to global internal tide typical scales and equation 5, respectively. Substituting
$\langle \xi_z^2 \rangle$ into equation (1) ultimately yields 32 m resolved vertical profiles of each observed profiles. The
dissipation rate $\varepsilon$ is related to the diapycnal diffusivity $K_z$ by the Osborn relation

$$K_z = \Gamma \frac{\varepsilon}{\overline{N^2}} \qquad (6)$$

where the flux coefficient $\Gamma$ is fixed at 0.2 generally.

**2.2  ERA-Interim and Slab-model**
The near-inertial energy flux for each observation profile was calculated using the 10 m wind speed
product from ERA-Interim (https://www.ecmwf.int/en /forecasts/datasets), which is 6-hourly wind
speed on a grid of $0.75° \times 0.75°$. We selected the mean near-inertial flux of 30-50 days before the
time of each diapycnal diffusivity estimation as our measure of the near-inertial flux, with the
consideration of the propagation of NIWs.
The wind-drive NIW energy flux can be directly estimated using a slab model, which assumes that





the inertial oscillations in the mixed layer do not interact with the background fields. The mixed layer
current velocity can be described by
$$\frac{dZ}{dt} + (r + if)Z = \frac{T}{\rho H} \tag{7}$$

where $Z = u + iv$ is the mixed layer oscillating component of full current, and $i$ is an imaginary
number to indicate the latitudinal component. $T = (\tau_x + i\tau_y)$ is the wind stress on the sea surface, $f$
is the local Coriolis parameter, $r$ is the frequency-dependent damping parameter, which was fixed
at $0.15\,f$ for these calculations. $\rho$ is sea water density and fixed at 1024 kg/m$^3$. $H$ is the mixed-layer
depth and was set to a constant 25 m. We can calculate the oscillating component of full velocity from
equation 7 and obtain the near-inertial component through a bandpass filter of $[0.85,\ 1.25]\,f$. The
near-inertial energy flux is calculated as
$$H(\Pi) = Re(Z \cdot T^*) \tag{8}$$

the asterisk (*) indicates the complex conjugate of a variable.

### 2.3 AVISO and Eddy kinetic energy

The eddy kinetic energy is estimated based on geostrophic calculation as:
$$EKE = \frac{1}{2}\left(U_g'^2 + V_g'^2\right) \tag{9}$$

$$U_g' = -\frac{g}{f}\frac{\Delta\eta'}{\Delta y} \qquad V_g' = -\frac{g}{f}\frac{\Delta\eta'}{\Delta x} \tag{10}$$

where $U_g'$ and $V_g'$ are the geostrophic velocities in the east-west and north-south directions,
respectively. They are taken from the AVISO (http://www.aviso.altimetry.fr/duacs/) geostrophic
velocity product. $\eta'$ indicates sea level anomaly (SLA).

### 2.4 Internal tidal conversion rates

The internal tidal conversion rate was provided by SEANOE (https://www.seanoe.org/data/, C.de
Lavergne et al., 2019), including 8 main tidal constituents. We used the mode-summed internal tidal
conversion rates of $M_2$ and $K_1$, and integrated 8 main tidal constituents in present study.

### 3.  Results

### 3.1 Spatial pattern of diapycnal mixing in the upper Philippine Sea

The diapycnal diffusivities were used as indicators of ocean diapycnal mixing. The pattern averaged
within 250-500 m is shown in Fig.1a. The $K_z$ was estimated from the ARGO profiles, with an average





on each cell of 0.5°×0.5°. The magnitude of diapycnal diffusivities increased with latitude, reaching
$10^{-4} m^2 s^{-1}$ in the northern part of this area (30°N-36°N). The mean value of $K_z$ was about O(-6)-O(-5)
in lower latitude. While, it was remarkable that the magnitude of $K_z$ also increased significantly in
some low-latitude regions, reaching O(-4) or higher. These regions include Izu Ridge (Nagasawa et al.,
2005; Tanaka et al., 2018) and Luzon Strait. Reviewing the influence of topography, wind and internal
tide (Fig.1b-d) on ocean mixing, it was found that the zonal variability of $K_z$ was consistent with the
wind intensity distribution. Upper ocean mixing was significantly enhanced at mid-latitudes due to the
presence of westerlies. In addition, $K_z$ was also enhanced near several key internal tide sources, such
as the Luzon Strait, Bonin Ridge, Izu Ridge, Dadong Ridge, etc. At these sites, the magnitude of $K_z$
was obviously larger than other areas at the same latitude, indicating a significant role of internal tides.
Additionally, the enhancement of deep ocean mixing at these sites was even more obvious (not shown).

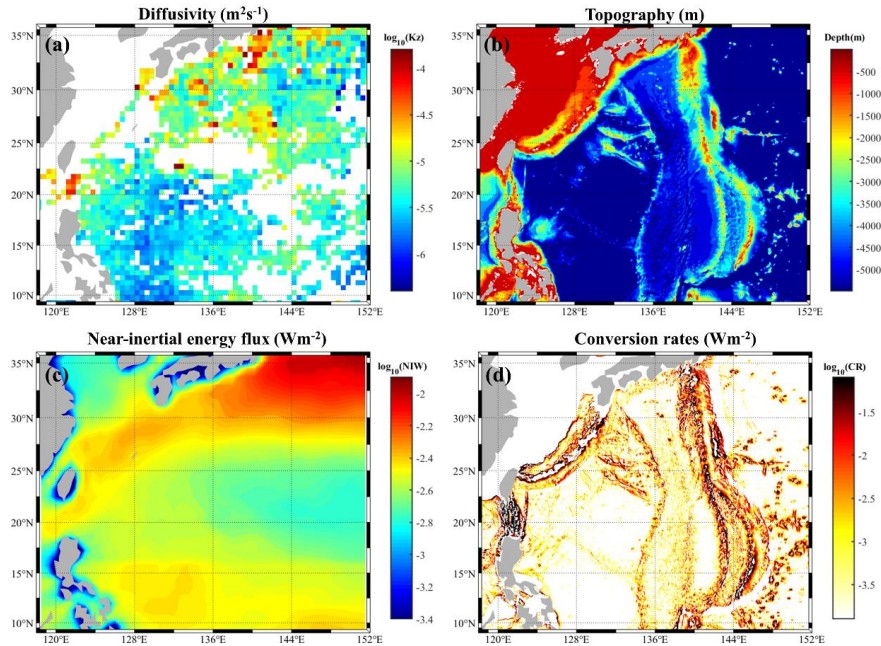

**Figure 1  Maps of (a) log-scale averaged diapycnal diffusivities $K_z$ (m²s⁻¹) estimated from ARGO profiles,**
**(b) topography, (c) log-scale long-term averaged near-inertial energy flux from wind (Wm⁻²), and (d)**
**log-scale M2 internal tide conversion rates (Wm⁻²).**

It can be noted that the pattern of diapycnal diffusivities was not completely consistent with those of
either internal tides or winds. This suggests that the ocean mixing was modulated by other factors than
tides and winds. The magnitudes of $K_z$ also vary for internal tide source sites. Considering that the
Philippine Sea is a region with energetic mesoscale motions (Fig.2), the influences of mesoscale
features in turbulent mixing should be taken into account. The existence of mesoscale features can alter
the propagation and dissipation of internal tides. Therefore, the Philippine Sea is an ideal region to
study the modulation of background flows on turbulent mixing associated with strong internal tides.

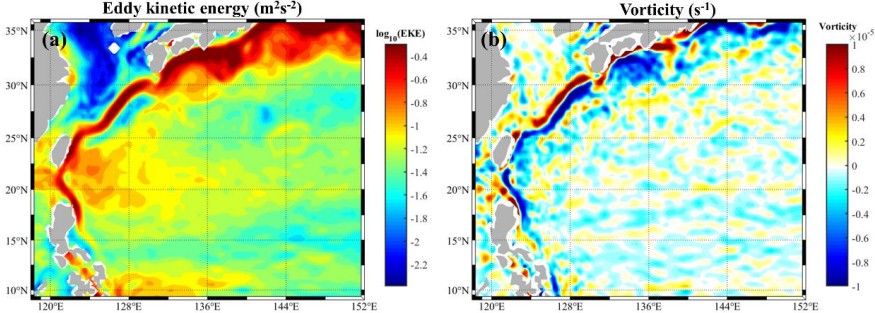


**Figure 2 Maps of (a) log-scale long-term averaged eddy kinetic energy and (b) long-term averaged vorticity.**

### 3.2  Seasonal variability of mixing at different latitudes

The seasonal cycle for diapycnal diffusivities also differs zonally. Here, we divided the Philippine

Sea into two portions: low-latitude (10 °N-25 °N) and mid-latitude (25 °N-35 °N). The diapycnal
diffusivities $K_z$ were averaged in each latitude band (Fig.3). At the depth of 250-500 m in the
mid-latitude, the diapycnal diffusivities had a significant seasonal trend as strong in winter and weak in
summer. This is consistent with the seasonal fluctuation of near-inertial energy from wind. Such a
seasonal cycle could also be found at 500-1000 m and 1000-1500 m in the mid-latitudes, but it was
relatively weaker, especially after 2016. In the lower latitudes, the NIW energy was still strong in
winter and weak in summer, but a seasonal dependence of turbulent mixing was not obvious, even in
the upper ocean. Consequently, the wind was found to play a significant role in driving turbulent
mixing at mid-latitude, but was insignificant at low latitudes. Other factors drove and modulated
turbulent mixing in low latitudes.

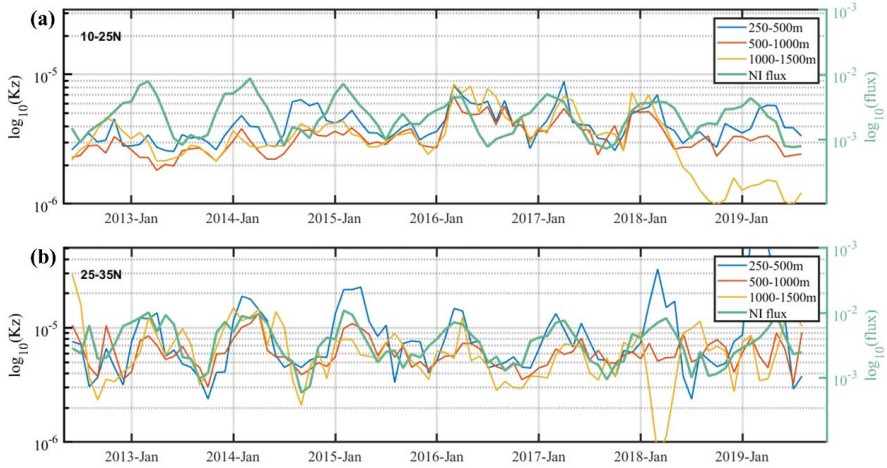


**Figure 3 Seasonal cycles in diapycnal diffusivities (colorful line) and near-inertial energy flux from wind**
**(green) extents to 250-500 m, 500-1000 m and 1000-1500 m in (a) 10 N -25 N and (b) 10 N-25 N, which is**
**averaged in each month and all water column.**


**3.3 Impact factors**

**3.3.1 Relative contributions**

The turbulent mixing of the Philippine Sea displayed an obvious zonal dependence, so the latitudinal

influence was examined for several factors: internal tides, wind and eddy kinetic energy (Fig.4). The

rates of diapycnal diffusivities in regions of weak/strong internal tides, weak/strong NIW energy, and

high/low eddy kinetic energy were calculated in each 1 °latitude, respectively. If the rate was close to 1,

the influence of this factor was insignificant, while a larger rate indicated a greater contribution.


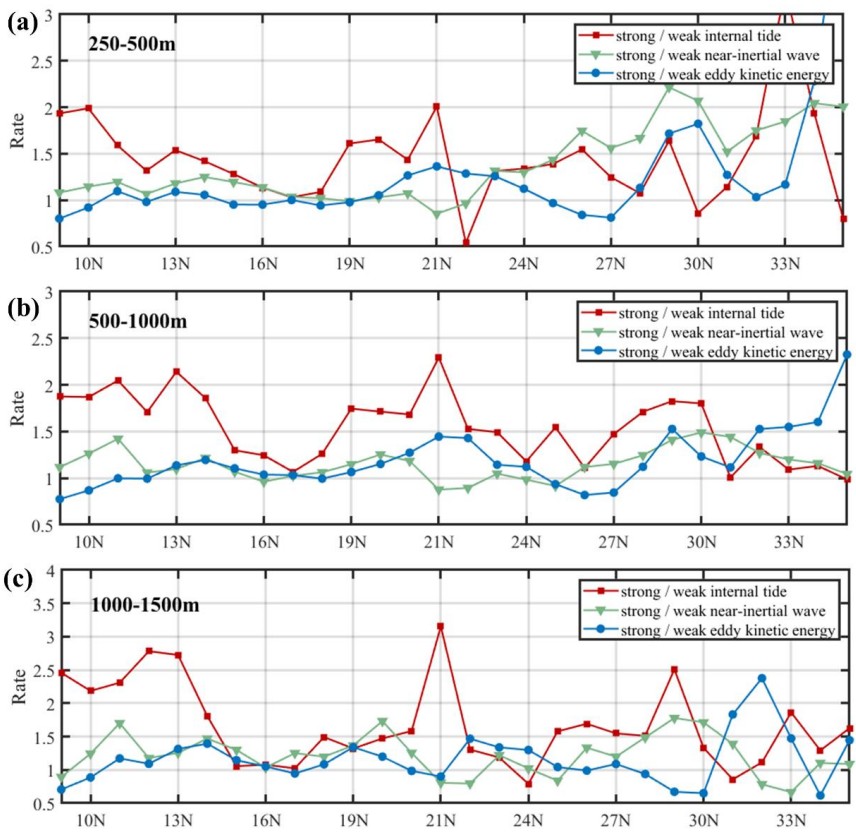


**Figure 4 Rates of diapycnal diffusivities between areas over strong (greater than median) and weak internal**
**tide (red lines), strong (greater than median) and weak near-inertial wave (green lines) and strong (greater**
**than median) and weak eddy kinetic energy (blue lines) for each 1 °latitude bands in the depth range of (a)**
**250-500 m, (b) 500-1000 m and (c) 1000-1500 m, Which averages for each bands containing more than 10**
**estimates.**


At depths of 250-500 m, the rate associated with internal tide increased significantly at 10 °N, 21 °N
and 33 °N. These latitudes correspond to Guap seamount, Luzon Strait and Izu Ridge, which are main
internal tide source sites. It reached 2 near these three latitudes, indicating that strong internal tides
triggered the enhancement of $K_z$ twice as much compared to the regions of weak internal tides. In
addition, north of 23 °N, the rate in related to NIW in the upper ocean increased with latitude. And it
exceeded the internal tidal contribution north of 25 °N, which indicated that the wind plays a more
important role in mixing at this latitude band. Taking the wind as the driving factor better explains the
seasonal cycle of diapycnal diffusivities in Fig.3, since the winds have an apparent seasonal





dependence. The obvious seasonal trend of $K_z$ due to the important contribution of wind occurs
between 25 °N-35 °N. In contrast, the rate for wind is ~1 at lower latitudes, indicating that the
wind-driven mixing is insignificant here with the absence of wind-driven seasonal cycle.
The contribution of wind to turbulent mixing is significantly reduced in the depth ranges of
500-1000 m and 1000-1500 m (Fig.4 b and c). The rate only increased slightly at mid-latitudes, less
than 2 anywhere. In contrast, the enhancement of mixing triggered by internal tides at these depth
ranges was more significant, with the rates exceeding 3.5 at some latitudes. This suggested that internal
tides played a more important role in deep ocean mixing. Furthermore, internal tides significantly
enhanced $K_z$ around 13 °N, 21 °N, and 29 °N, corresponding to the sources of Mariana Trench, Luzon
Strait and Bonin Ridge, respectively. Such enhancement was not obvious at the Izu Ridge possibly due
to the shallower depth and paucity of deep data, or the turning latitude effects in this area.
Combined with the analysis of relative contributions of different factors in different layers, it was
concluded that the contribution of internal tides in turbulent mixing is more important in low latitudes
of the Philippine Sea. In this area, the wind and mesoscale features did not significantly enhance $K_z$. At
mid-latitudes, internal tides still played an important role, but the wind contribution was more
significant in the upper ocean. The wind drove turbulent mixing even at the depths of 500-1000 m and
1000-1500 m. The mid-latitude region not only corresponds to westerlies, but also features energetic
mesoscale motions. Therefore, the mesoscale features might be a potential factor for the enhanced
turbulent mixing.
In the low latitudes, $K_z$ did not increase in the regions of high eddy kinetic energy or strong
near-inertial energy, whereas, it increased significantly in the regions of strong internal tides. This
enhancement was more obvious below 400 m (Fig.5 a). And in the mid-latitudes, $K_z$ in the upper
ocean increased significantly corresponding to strong winds with compared to weak winds (Fig.5 b).
Meanwhile, $K_z$ was also larger in the regions of strong internal tides and high EKE in the upper ocean.
The enhancement of wind or EKE to turbulent mixing significantly weakened below 600 m, while the
enhancement of internal tides increased with depth. This indicates that internal tides played a
significant role in turbulent mixing not only in the low latitudes, but also in the mid latitudes with their
strong winds.


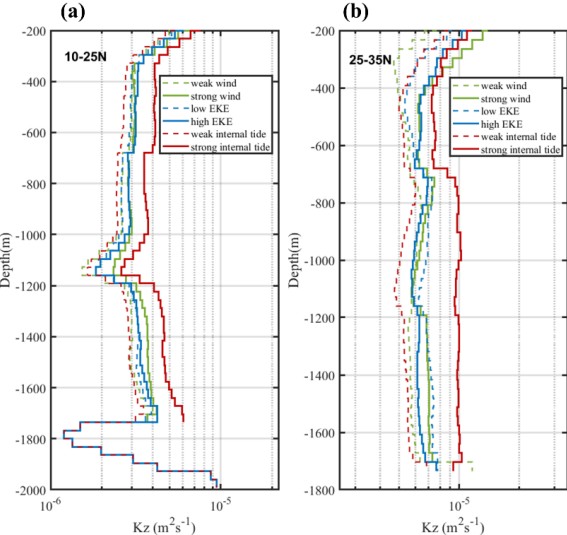

**Figure 5 Vertical structures of geometric averaged diapycnal diffusivities $K_z$ with weak and strong wind (green), low and high EKE (blue) and weak and strong internal tide (red) in the (a) low-latitude and (b) middle latitudes.**

### 3.3.2 Wind

The least squares linear fit slopes of $K_z$ to NIW energy from wind represent the mixing response to wind. Here, the Philippine Sea is divided into 10 °N -15 °N, 15 °N-25 °N and 25 °N-35 °N (Fig.6). At the depth of 250-500 m, the slope is the largest in 25 °N-35 °N (~0.305), followed by that in 10 °N -15 °N (~0.133), and the smallest in 15 °N-25 °N (~0.013). The wind driven turbulent mixing was most significant between 25 °N-35 °N, but was insignificant between 15 °N-25 °N. At the depth of 500-1000 m, the wind influence on turbulent mixing was weakened in the mid-latitudes. This was consistent with the results of Fig.3 and Fig.4. It proved that the contribution of wind has a zonal dependence, which was significant at the mid-latitudes, but insignificant at low latitudes. In addition, the response of turbulent mixing to wind weakened quickly with depth, indicating that the dominant factor of mixing in the deeper water column was not wind. Accordingly, it was difficult for wind to drive mixing below 1000 m, so we do not show the results at the depth of 1000-1500 m (Fig.4).



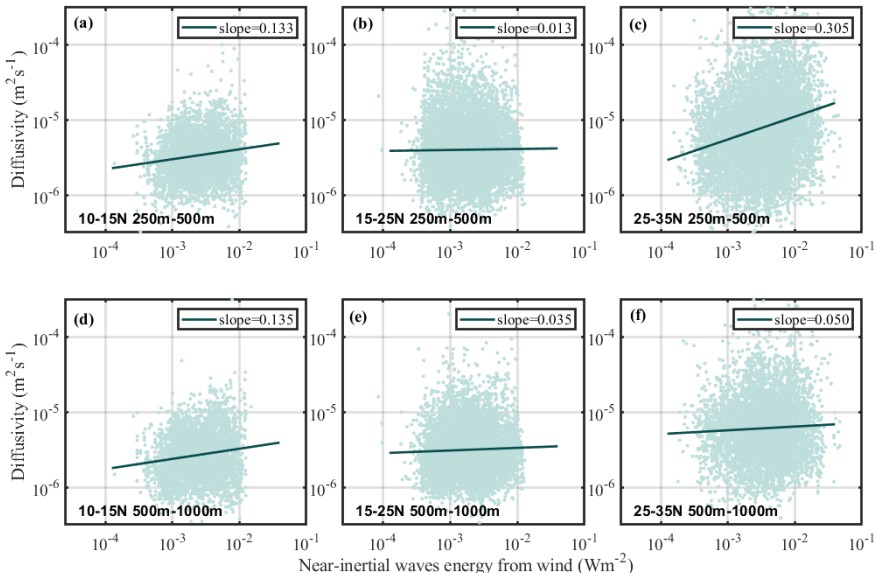

**Figure 6 Scatter of log-scale Kz versus log-scale near-inertial energy flux from wind in 250-500 m between (a)**

**10 °N -15 °N, (b) 15 °N-25 °N and (c) 25 °N-35 °N, and in 500-1000 m between (d) 10 °N-15 °N, (e) 15 °N-25 °N and**

**(f) 25 °N-35 °N The best-fit slopes are denoted by the solid line.**

### 3.3.3 Tide

The slopes of $K_z$ to internal tides conversion rates represent the mixing response to internal tides. As

discussed above, the mixing significantly responded to the internal tides over the entire Philippine Sea

(Fig.7). The relationship was depth dependent. The slopes did not reach 0.1 at the depth of 250-500 m,

but increased significantly at 500-1000 m and 1000-1500 m. and exceed 0.13 for the deepest depth

band. The response of mixing to internal tides was more significant in the deeper ocean. Focusing on

different latitude bands, the slopes of $K_z$ to internal tides is smaller at mid-latitudes. This is because

the wind contribution increased in this region, which led to a weakening relative contribution of

internal tides. Compared with the internal tide conversion rates, the pattern of $K_z$ was inconsistent

with internal tides, even at lower latitudes. It can be inferred that the turbulent mixing was not only

affected by the internal tides, but also by other factors. There is a strong western boundary flow,

Kuroshio, and an active mesoscale environment in this region. Some researchers have shown that the

existence of mesoscale environment will alter the internal tide features, so we reasonably infer that the

tidal induced turbulent mixing in this area was modulated by the mesoscale features.

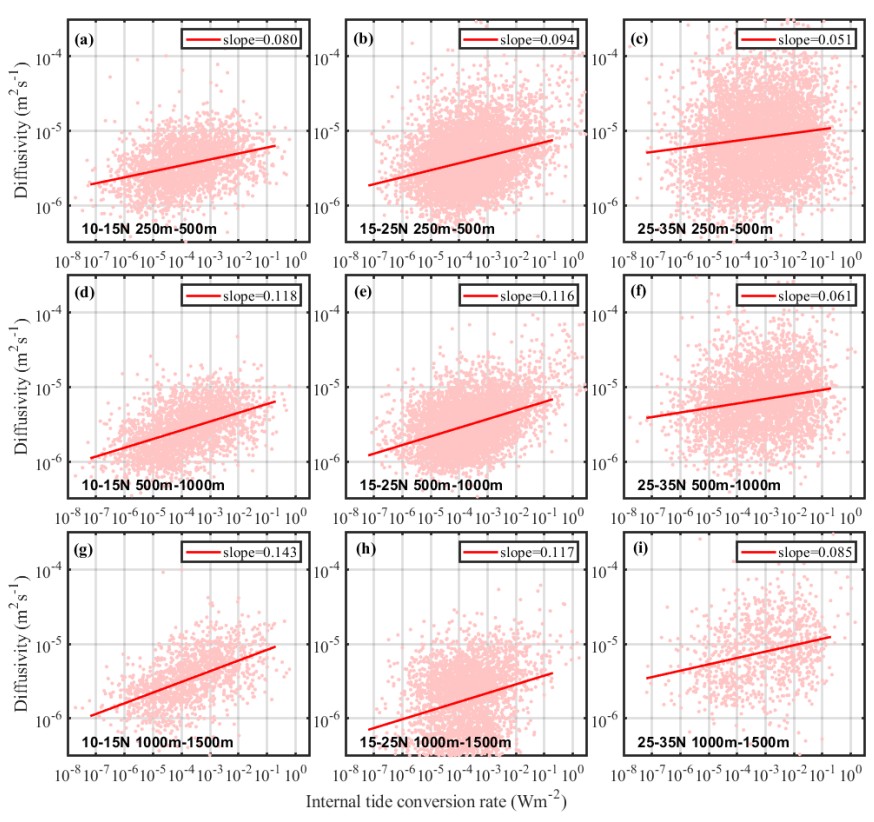

**Figure 7 Scatter of log-scale $K_z$ versus log-scale internal tide conversion rate in 250-500 m (row 1), 500-1000 m (row 2), 1000-15000 m (row 3) and the best-fit slopes are denoted by the red line. Columns 1,2,3 are 10 °N-15 °N, 15 °N -25 °N and 25 °N -35 °N latitude bands, respectively.**

### 3.4 Role of Mesoscale features in tidal mixing

Focusing on the low latitudes, where tidal mixing is dominated, the diapycnal diffusivities, $K_z$, related to internal tides and eddy kinetic energy are shown (Fig.8). The combined influences of mesoscale features and internal tides on mixing are indicated. The increasing internal tide conversion rates significantly enhanced turbulent mixing. When the conversion rate was $10^{-3} Wm^{-2}$, the magnitudes of $K_z$ were about $3 \times 10^{-6} m^2 s^{-1}$, $3 \times 10^{-6} m^2 s^{-1}$, $1 \times 10^{-6} m^2 s^{-1}$ at the depth of 250-500 m, 500-1000 m and 1000-1500 m, respectively. When the internal tide conversion rates reached O(-1)-O(0), $K_z$ reached $10^{-5} m^2 s^{-1}$ at both depths of 250-500 m and 500-1000 m, even exceed $10^{-4} m^2 s^{-1}$ at some internal tide source sites. In addition, there was a positive correlation between eddy kinetic energy and





diapycnal diffusivites. A higher eddy kinetic energy can further increase $K_z$ under the same magnitude
of internal tide conversion rate. Such enhancement was more significant with strong internal tide
conversion rates greater than $10^{-3}\,Wm^{-2}$.

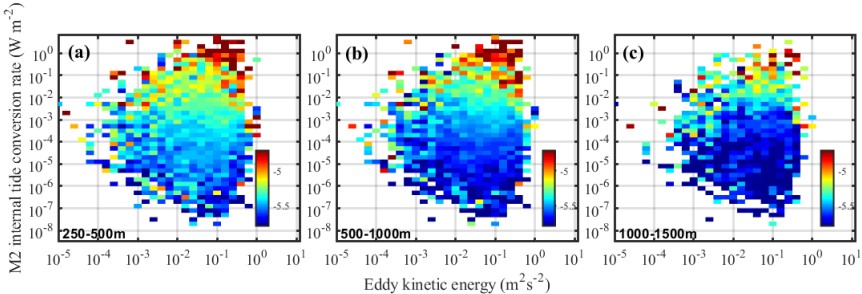


**350**     **Figure 8 Averaged diapycnal diffusivities as a function of EKE and internal tide conversion rates between (a)**

**351**     **250-500 m, (b) 350-500 m and (c) 500-1000 m.**


$M_2$ and $K_1$ tidal constituents were analyzed to clarify the response of $K_z$ to internal tides in the
regions of high eddy kinetic energy (EKE is larger than the regional average value) and low eddy
kinetic energy (Fig.9). The results integrating 8 main tidal constituents (Fig.9 a, b and c) showed that
the slopes in a weak (strong) mesoscale field were smaller (larger), 0.081 (0.105), 0.103(0.134) and
0.103 (0.142) at the depth of 250-500 m, 500-1000 m, 1000-1500 m, respectively. The turbulent mixing
was more sensitive to the internal tide magnitude in the presence of an energetic mesoscale field.
Moreover, such response was more obvious in the region with strong internal tides (such as $>10^{-2}Wm^{-2}$
*conversion rate*). In some regions with weak internal tides, such as with internal tide conversion rates
less than $10^{-3}Wm^{-2}$, the modulation of mesoscale eddies was less significant.





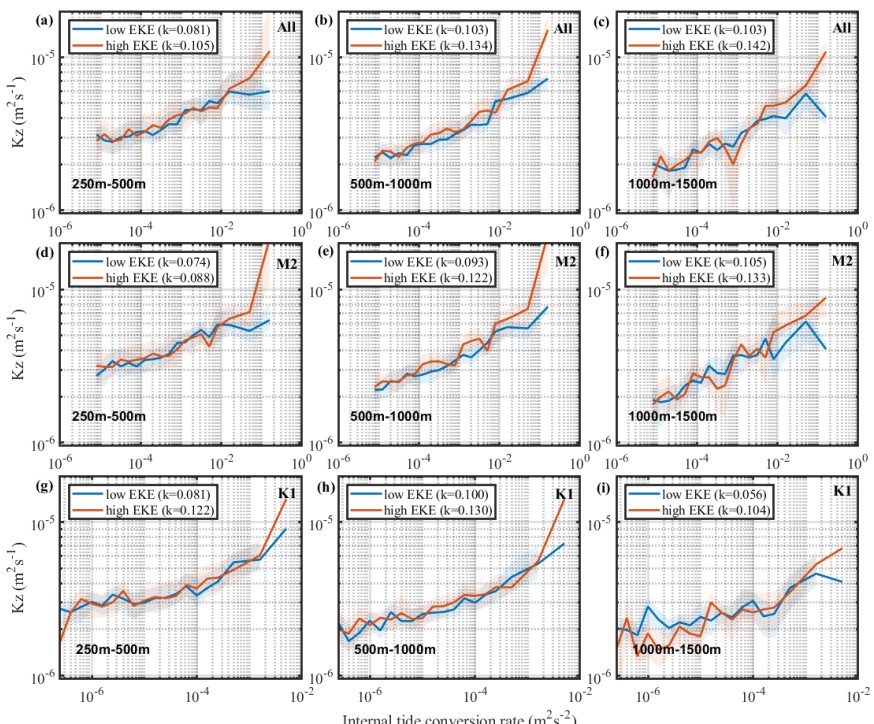

**Figure 9 The averaged diffusivity between depths of (a, d, g) 250 m-500 m, (b, e, h) 500 m-1000 m and (c, f, i) 1000 m-1500 m in high (greater than the median) and low (less than the median) eddy kinetic energy. The shade indicate the 1 deviation. Rows 1,2 and 3 are related to 8 main tidal constituents, $M_2$ internal tide and $K_1$ internal tide, respectively.**

A similar conclusion can be drawn only considering $M_2$ or $K_1$. In regions of high eddy kinetic energy, the change in diffusivities in response to internal tides was significant. And the increase was more sensitive to $M_2$ internal tide. The enhancement related to $M_2$ internal tide was more significant below 500 m (Fig.9 d and e), while enhancement of the $K_1$ internal tide was similar at all depths. This may be due to different features and structures of $M_2$ and $K_1$ internal tides. In this area, the modal structure and propagation path of $M_2$ internal tide are more complicated and more prone to breaking, but those of $K_1$ were relatively stable. And this area includes the $K_1$ critical latitude range, which can be broadened by mesoscale currents (Robertson and Dong, 2019).

The modulation of cyclonic and anticyclonic eddies on tidal mixing also differ. The increase of $K_z$ by internal tides in regions with cyclonic eddies (vorticity$>3 \times 10^{-6} s^{-1}$) and anticyclonic eddies (vorticity$<-3 \times 10^{-6} s^{-1}$) are both shown (Fig.10 and Fig.11). Under the same magnitude of internal tides,

the $K_z$ increase more significantly in the presence of anticyclonic eddies, which is obvious in 250-500
m, and can also be seen in 500-1000 m. Below 1000 m, there is no significant differences between the
regions with cyclones and anticyclones.

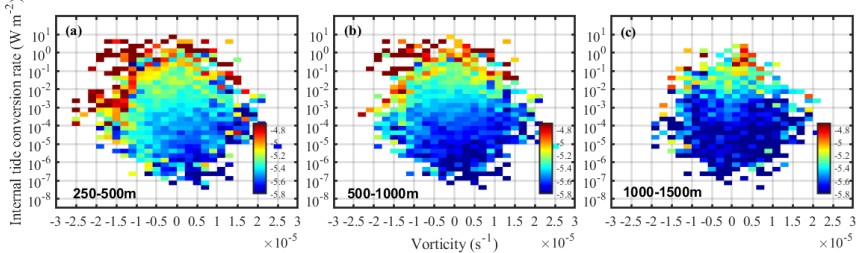

**Figure 10 The averaged diapycnal diffusivities as a function of vorticity and internal tides conversion rate**
**between (a) 250-500 m, (b) 500-1000 m and (c) 1000-1500 m**

Considering mixing driven by eddies is relatively significant in regions where the tidal mixing is

very weak, we only analyze the cases of internal tides conversion rates larger than $10^{-3} Wm^{-2}$. When the
conversion rates become larger than this value, the diapycnal diffusivities at the presence of high eddy
kinetic energy increase faster with internal tides (Fig. 9). It was found that the response of turbulent
mixing to internal tides was more sensitive in the presence of anticyclones above 1000 m. While below
1000 m, the influence of cyclones is slightly higher than that of anticyclones.

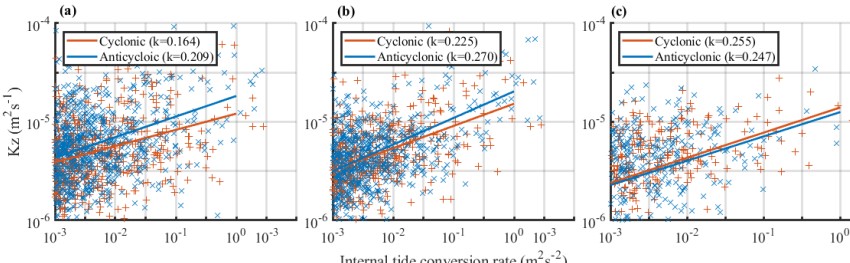

**Figure 11 scatter of log-scale $K_z$ versus log-scale internal tide conversion rate with Cyclone (red) and**
**anticyclone (blue) in (a) 250-500 m, (b) 500-1000 m, and (c) 1000-1500 m. The best-fit slopes are denoted by**
**the red and blue solid line.**
**4.   Summary and Discussion**

The spatial pattern and seasonal variability of the diapycnal diffusivities in the Philippine Sea were

estimated using a fine scale parameterization. The main conclusions are as follows.





The seasonal fluctuations of mixing in this area were zonally dependent. Seasonal variability was
strong in winter and weak in summer at mid-latitudes, with the seasonal fluctuations more obvious in
the upper ocean. This was attributed to the Westerlies, and the wind plays a more significant role in
turbulent mixing here. However, the seasonal cycle of mixing in the low latitudes was not obvious,
indicating that the wind-driven mixing was not dominant here. As opposed to wind-driven mixing, tidal
mixing was more significant in the deeper ocean.
Evidence that the mixing was modulated by internal tides was seen in regions of both high and low
eddy kinetic energy, and it was more significant with high eddy kinetic energy. The presence of high
eddy kinetic energy enhanced the response of $K_z$ to internal tides, especially for the $M_2$ internal tide.
The increased rate of $K_z$ with internal tides in the high EKE field was higher than that in the weak
EKE field. The existence of mesoscale features changed the vertical structure of internal tides, and
transferred the internal tides energy from low modes to higher modes. It was more likely to cause
internal tide breaking (Dunphy and Lamb 2014). The enhancement by mesoscale motions to tidal
mixing was more significant for $M_2$ internal tides. Anticyclonic eddies were more likely to increase
tidal mixing in the upper ocean. While the influence of cyclonic eddies to tidal mixing was slightly
higher than that of anticyclonic ones in the deep ocean.
There are several mechanisms that might explain the elevated tidal mixing in the present of energetic
mesoscale environment. The vertical scales of internal tide can be reduced and the energy of internal
tide can be amplified near the surface in the presence of energetic mesoscale features. When internal
tide passes through mesoscale eddy, the energy of mode-1 internal tide can be refracted and transmitted
to higher-mode waves (eg. Farrari and Wunsch, 2008, Henning and Vallis, 2004). The eddy flows can
also directly increase vertical shear and subsequently internal tide energy dissipation rate (eg.
Chavanne et al., 2010, Dunphy, 2014). The anticyclones induce higher tidal mixing than do cyclones
probably because of the Chimney effects associated with distinct vorticities (Jing and Wu, 2011).
This paper explores the modulation of the mesoscale environments on tide-induced mixing
statistically by some observed datasets. Theoretical clarification of the driving mechanisms is needed.
Some previous numerical studies can explain our conclusion to some extent. However, how and to
which extent the vorticity alter internal tide evolution and induced mixing have not been clearly
explained in theory. Moreover, the latitude range from 9 °N to 36 °N are discussed in this work due to
the limitation of fine scale parameterization method in equatorial areas. The influence of the equatorial



background flows on ocean mixing remains to be solved.

**Code and data availability.** The ARGO data (ftp://ftp.argo.org.cn/pub/ARGO/global/) set were
made available by China Argo Real-time Data Center (Li Zhaoqin et al., 2019). The near surface 10 m
wind speed was product by ERA-Interim dataset (https://www.ecmwf.int/en /forecasts/datasets). The
geostrophic velocity were taken from the AVISO (http://www.aviso.altimetry.fr/duacs/). The internal
tidal conversion rate was provided by SEANOE (https://www.seanoe.org/data/, C.de Lavergne et al.,
2019). The corresponding data and codes are available on request to Zhenhua Xu by email.

**Author contribution.** The concept of this study was developed by Zhenhua Xu and extended upon
by all involved. Jia You implemented the study and performed the analysis with guidance from
Zhenhua Xu, Qun Li and Robin Robertson. Peiwen Zhang and Baoshu Yin collaborated in discussing
the results and composing the manuscript.

**Competing interests.** The authors declare that they have no conflict of interest.

**Acknowledgment.** Funding for this study was provided by the Strategic Priority Research Program
of Chinese Academy of Sciences (No.XDB42000000, XDA11010204), the National Key Research and
Development Program of China (No. 2016YFC1402705, 2017YFA0604102), the National Natural
Science Foundation of China (91858103, 41676006), Key Research Program of Frontier Sciences,
CAS.

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
