# Peer review of "Enhanced internal tidal mixing in the Philippine Sea"

_Nonlinear Processes in Geophysics, 2021_

## Author Comment (AC1)

**Referee #1**

The paper deals with an interesting and important problem, namely the mechanism(s) responsible for oceanic mixing in a topographically-complex region. The data have been processed satisfactorily and they show some interesting features, though the interpretation of these features is not straightforward. My principal difficulty with the paper lies in the analysis of the data and the validity of the conclusions drawn from such an analysis. The heart of the analysis is the correlation of observed features of the diapycnal diffusivity patterns with the main causes of mixing identified by the authors. Many of the correlations claimed by the authors seem to be based either on visual inspection of the data plots or analysis based upon ratios of the energy fluxes associated with each of the driving agencies (e.g Fig 4) . I did not find these analyses convincing and I would have appreciated some more quantitative and rigorous data correlation procedures carried out to justify the conclusions. Figs 6, 7 and 8 are very difficult to interpret. Overall, I found the data analysis to be rather superficial and, in consequence, the conclusions unjustified by the evidence of the data analysis.

*Response:*

*Thank you for your time and constructive comments. The convincing verifications of both the methods and results are crucial for a scientific paper. According to your helpful suggestions, we have thoroughly made the validation of the analysis methods used in present manuscript. The detailed descriptions and discussions were added to section 3.3.1 (Fig. 4 and Fig. 5), section 3.3.2 (Fig. 6 and Fig. 7) and section 3.4 (Fig. 8). Based on the verified methods and the comparison to other studies, the conclusions that have been drawn are found to be reliable and convincing.*

*1. As for the parameterization method itself, it is widely used in the studies of ocean mixing. The most beneficial point is that there are amount of observations of temperature and salinity by ARGO among the global ocean, with good spatial and temporal coverage. The parameterization method was used to estimate global or regional distributions of diapycnal mixing, which can supplement the scarcity of direct turbulence observations. The parameterization analysis seems not straightforward but effective in mixing investigation through several statistical analysis and correlation analysis with other impact factors (e.g. Wu et al., 2011; Whalen et al., 2012; Kunze et al., 2017; Chanona et al., 2018).*

*2. As for the statistical and correlation analysis methods*

*a. To address the relative contributions to the mixing from possible factors, we showed Fig. 4 and Fig. 5. Each 1° latitude band was separated into two regions with weak or strong internal tides (or other factors). Here we define the strong or weak internal tides as the internal tide conversion rate larger or smaller than the median internal tide conversion rate of the Philippine Sea (for the details see section 3.3.1). The diapycnal diffusivities in these two kinds of regions were then averaged. The similar method has been used to analyze the effect of topography and different frequency-bands internal waves on ocean interior shear and mixing (eg. Whalen et al., 2012; Zhang et al., 2018). For a more direct representation, the ratios of diapycnal diffusivities above the strong internal tides to weak internal tides were shown. The ratio larger than 1 means that the diapycnal diffusivities are significantly higher in the regions of strong internal tides. The similar analysis were used for the analysis of wind and EKE effects.*

[Figure]

**Figure 4 Ratios of diapycnal diffusivities between areas over strong (greater than median) and weak internal tide (red lines), strong (greater than median) and weak near-inertial wave (green lines) and strong (greater than median) and weak eddy kinetic energy (blue lines) for each 1° latitude bands in the depth range of (a) 250-500 m, (b) 500-1000 m and (c) 1000-1500 m, Which averages for each bands containing more than 10 estimates.**

[Figure]

**(Whalen et al., GRL, 2012, Figure 3a ) E.g. Global mean dissipation rate for 3_x0005_ half-overlapping latitudinal bands in the depth range 250–1,000 m over rough (variance greater than global mean) and smooth topography with 90% bootstrapped confidence intervals.**

[Figure]

**Figure 5 Vertical structures of geometric averaged diapycnal diffusivities $K_z$ with weak and strong wind (green), low and high EKE (blue) and weak and strong internal tide (red) in the (a) low-latitude and (b) middle latitudes.**

[Figure]

(Chanona et al., JGR, 2018, Figure 11 ) E.g. (a) Spatial distribution of strong (purple) and weak (blue) barotropic tidal speeds (b) spatial distribution of rough (purple) and smooth (blue) topography as determined by topographic roughness, and (c) log-scale histogram of stormy (purple) and calm (blue) 10 m wind speeds, associated with each conductivity-temperature-depth profile. High/low bin cutoffs for each parameter are defined by the upper/lower quartile limits of their corresponding distributions; gray regions indicate interquartile values not included in our analysis. Average vertical profifiles of $\varepsilon_{IW}$ (middle column) and $K_{IW}$ (right column) are binned accordingly for (d, g) strong/weak tidal speeds, (e, h) rough/smooth topography, and (f, i) stormy/calm wind speeds. The number of profiles comprising each vertical average is given by n and shading indicates the associated geometric standard deviations.

*b. In order to further verify our results of Fig. 4 and. 5, the Fig. 6 and Fig. 7 were plotted using another direct correlation analysis method. The same conclusion can be drawn from the two kinds of analysis methods. The linear regression is a popular method for statistics analysis of the correlation between two factors (eg. Wu et al., 2011; Jing et al., 2014; Jeon et al., 2018; Zhao et al., 2019). The regression coefficient can represent the response of mixing to wind (eg. Qiu et al., 2012) or other factors. Another reviewer also gave some advises for comparing and interpreting these figures. Noted that Fig. 6 and Fig. 7 have been revised according to the suggestions from both reviewers.*

[Figure]

**Figure 6 Scatter of log-scale Kz versus log-scale near-inertial energy flux from wind in 250-500 m between (a) 10°N -25°N and (b) 25°N-35°N, and in 500-1000 m between (c) 10°N-25°N and (d) 25°N-35°N The best-fit slopes are denoted by the solid line, the 95% confidence interval is indicated by dash lines.**

[Figure]

**Figure 7 Scatter of log-scale $K_z$ versus log-scale internal tide conversion rate in 250-500 m (row 1), 500-1000 m (row 2), 1000-15000 m (row 3) and the best-fit slopes are denoted by the red line. Columns 1 and are 10°N-25°N and 25°N -35°N latitude bands, the 95% confidence interval is indicated by dash lines.**

*3. As for the conclusions*

*The main conclusions in this paper include: 1. internal tides played a significant role in mixing the whole water column in the Philippine Sea. 2. the energetic mesoscale environment increases the internal tidal mixing.*

*We believe our conclusions are reliable and convincing, with several reasons:*

*a. The methods used are reasonable. The statistical methods have all passed the significance test.*

*b. We used different methods for the analysis and come to the same conclusion (eg. Figs. 4,5 and Figs. 6,7; Fig. 8 and Fig. 9). The results from different methods can support each other.*

*c. Some conclusions can be supported by previous studies. For example, according to Fig. 4 and Fig. 5, we can get the following three messages: 1. Wind and EKE play important roles on mixing the upper ocean in the mid-latitudes; 2. Strong internal tides are easier to enhance mixing in the deeper ocean. 3. Internal tides played a significant role in turbulent mixing not only in the low latitudes, but also in the mid latitudes with r strong winds. The points 1 and 2 are consistent with some previous works (e.g. Waterhouse et al., 2014; Mackinnon et al., 2017; Whalen et al., 2018). The point 3 can be explained by some theoretical and numerical studies. For example, the eddy flows can increase vertical shear and subsequently internal tide energy dissipation rate (eg. Chavanne et al., 2010, Dunphy, 2014).*

*Therefore, we carefully considered your comments, checked and verified our methods and results again. We believe our methods used are valid and our main conclusions are reliable and reasonable.*

There are a few typographic and/or grammatical errors but, in general, the standard of English is satisfactory. The main error is to use "rates" instead of "ratios" in the correlation analyses (line 250 and elsewhere). The use of "slopes" instead of "ratios" (e.g line 319 and elsewhere) is misleading also (if I have understood the text correctly). In equation (3) I assume that "acrcosh" should be "arccosh"?

**Response:**

*Thank you for your comments, the "rates" (line 250 and elsewhere) was misused in this paper, and it has been corrected. The Figs. 7 and 8 used the linear regression approach, so we think the "slopes" is more suitable here. And the error in equation (3) has been corrected.*

[revised manuscript text omitted]

 Each 1°latitude band was separated into two regions with weak or strong internal tides (or other factors). Here we define the strong or weak internal tides as the internal tide conversion rate larger or smaller than the median of the Philippine Sea. The diapycnal diffusivities in these two kinds of regions were then averaged. The similar method has been used to analyze the effect of topography and different frequency-bands internal waves on ocean interior shear and mixing (eg. Whalen et al., 2012; Zhang et al., 2019). For a more direct representation, the ratios of diapycnal diffusivities above the strong internal tides to weak internal tides were shown. The ratio larger than 1 means that the diapycnal diffusivities are significantly higher in the regions of strong internal tides.This ratio larger than 1 means that the diapycnal diffusivities is higher in the regions of strong internal tides than above weak internal tide in this latitudinal band. The larger this ratio is ,the more important the internal tideal induced mixing. Similarly, the contributions of near-inertial wave and eddy kinetic energy arewere analyzed inby this statistical method. The strong near-inertial wave or eddy kinetic energy is the locations where this parameter exceeds the regional median.

The rates of diapycnal diffusivities in regions of weak/strong internal tides, weak/strong NIW energy, and high/low eddy kinetic energy were calculated in each 1° latitude, respectively. If the rate was close to 1, the influence of this factor was insignificant, while a larger rate indicated a greater contribution.

[Figure]

**Figure 4 Ratetios of diapycnal diffusivities between areas over strong (greater than median) and weak internal tide (red lines), strong (greater than median) and weak near-inertial wave (green lines) and strong (greater than median) and weak eddy kinetic energy (blue lines) for each 1° latitude bands in the depth range of (a) 250-500 m, (b) 500-1000 m and (c) 1000-1500 m, Which averages for each bands containing more than 10 estimates.**

277  At depths of 250-500 m, the ratio associated with internal tide increased significantly at 10°N,

278 21°N and 33°N. These latitudes correspond to Guap seamount, Luzon Strait and Izu Ridge, which are

279 main internal tide source sites. It reached 2 near these three latitudes, indicating that strong internal

280 tides triggered the enhancement of $K_z$ twice as much compared to the regions of weak internal tides. In

281 addition, north of 23°N, the ratio in related to NIW in the upper ocean increased significantly with

282 latitude.  which indicated that the wind

283 plays a more important role in mixing at this latitude band. This result is basically consistent with

284 previous studies (Whalen et al., 2018), which suggested that the mixing is dominated by wind in the

285 mid-latitude. Taking the wind as the driving factor better explains the seasonal cycle of diapycnal

286 diffusivities in Fig.3, since the winds have an apparent seasonal dependence. The obvious seasonal trend

287 of $K_z$ due to the important contribution of wind occurs between 25°N-35°N. In contrast, the ratio

288 for wind is ~1 at lower latitudes, indicating that the wind-driven mixing is insignificant here with the

289 absence of wind-driven seasonal cycle.

290  The  wind contribution to turbulent mixing is significantly reduced in the depth ranges

291 of 500-1000 m and 1000-1500 m (Fig.4 b and c). The ratio only increased slightly at mid-latitudes,

292 less than 2 anywhere. In contrast, the enhancement of mixing triggered by internal tides at these depth

293 ranges was more significant, with the ratios exceeding 3.5 at some latitudes. This suggested that

294 internal tides played a more important role in deep ocean mixing. Furthermore, internal tides significantly

295 enhanced $K_z$ around 13°N, 21°N, and 29°N, corresponding to the sources of Mariana Trench, Luzon

296 Strait and Bonin Ridge, respectively. Such enhancement was not obvious at the Izu Ridge possibly due

297 to the shallower depth and paucity of deep data, or the turning latitude effects in this area.

298  Combined with the analysis of relative contributions of different factors in different layers, it was

299 concluded that the contribution of internal tides in turbulent mixing is more important in low latitudes of

300 the Philippine Sea. In this area, the wind and mesoscale features did not significantly enhance $K_z$. At

301 mid-latitudes, internal tides still played an important role, but the wind contribution was more significant

302 in the upper ocean. The wind drove turbulent mixing even at the depths of 500-1000 m and 1000-1500

303 m. The mid-latitude region not only corresponds to westerlies, but also features energetic mesoscale

304 motions. Therefore, the mesoscale features might be a potential factor for the enhanced turbulent mixing.

305 The modulation of mesoscale environments in the wind-induced mixing are discussed by some previous

306 studies (eg. Jing et al., 2011; Whalen et al., 2018), while the impact of mesoscale features in tide-induced mixing and in lower latitudes were not be considered.

The Philippine Sea was separated into two latitude bands. The vertical structures of diapycnal diffusivities in the regions with strong or weak internal tides were compared (Fig.5). This result can directly reveal the enhancement of internal tide on mixing at different depths. A similar analysis was used for wind and EKE. In the low latitudes, $K_z$ did not increase in the regions of high eddy kinetic energy or strong near-inertial energy, whereas, it increased significantly in the regions of strong internal tides. This enhancement was more obvious below 400 m (Fig.5 a). And in the mid-latitudes, $K_z$

in the upper ocean increased significantly corresponding to strong winds with compared to weak winds (Fig.5 b). Meanwhile, $K_z$ was also larger in the regions of strong internal tides and high EKE in the upper ocean. The enhancement of wind or EKE to turbulent mixing significantly weakened below 600

m, while the enhancement of internal tides increased with depth. Here, these results convey several information: 1. Wind and EKE play important roles on mixing in the upper ocean in the middle latitudes; 2. Strong internal tidefacilitate and enhance mixing in the deeper ocean. These two conclusions are consistent with previous researchers (eg. Jing et al., 2011; Whalen et al., 2012;

Waterhouse et al., 2014; Mackinnon et al., 2017; Whalen et al., 2018). In addition, our results indicate that, in the Philippine Sea, internal tides play a significant role in turbulent mixing not only in the low latitudes, but also in the mid latitudes,and not only in the deeper ocean, but also in the upper ocean.

[Figure]

**Figure 5 Vertical structures of geometric averaged diapycnal diffusivities $K_z$ with weak and strong wind (green), low and high EKE (blue) and weak and strong internal tide (red) in the (a) low-latitude and (b) middle latitudes.**

**3.3.2 Wind**

We adopted the linear regression approach and obtained the correlation between diapycnal diffusivities and wind. This approach is generally used to statistics the correlation between two factors (eg. Wu et al., 2011; Jing et al., 2014; Jeon et al., 2018; Zhao et al., 2019). The regression coefficient is able to  represent the mixing response to wind (eg. Qiu et al., 2012). Here, the Philippine Sea is divided into 10°-25°N and 25°N-35°N (Fig.6). At the depth of 250-500 m, the slope is significant larget in 25°N-35°N (~0.305), ~~followed by that in 10°N  15°N (~0.133),theest5~~0°N-25°N (~0.029). The wind driven turbulent mixing was most significant between 25 °N-35 °N, but was insignificant between 10°N-25°N. At the depth of 500-1000 m, the wind influence on turbulent mixing was weakened in the mid-latitudes. This was consistent with the results of Fig.3 and Fig.4. It proved that the contribution of wind has a latitudinal dependence, which was significant at the mid-latitudes, but insignificant at low latitudes. In addition, the response of turbulent mixing to wind weakened quickly with depth, indicating that the dominant factor of mixing in the deeper water column was not wind. Accordingly, it was difficult for wind to drive mixing below 1000 m, so we do not show the results at the depth of 1000-1500 m (Fig.4).

[Figure]

**Figure 6 Scatter of log-scale Kz versus log-scale near-inertial energy flux from wind in 250-500 m between (a)**
**10°N -2°N  and (e) 25°N-35°N, and in 500-1000 m between (d) 10°N-2°N **
**and (f) 25°N-35°N The best-fit slopes are denoted by the solid line, the 95% confidence interval is indicated**
**by dash lines.**

**3.3.3 Tide**

The slopes of $K_z$ to internal tidal conversion rates represent the mixing response to internal tides. As discussed above, the mixing significantly responded to the internal tides over the entire Philippine Sea (Fig.7). The relationship was depth dependent. The slopes did not reach 0.1 at the depth of 250-500 m, but increased significantly at 500-1000 m and 1000-1500 m. and reach 0.18 for the deepest depth band. The response of mixing to internal tides was more significant in the deeper ocean. Focusing on different latitude bands, the slopes of $K_z$ to internal tides is smaller at mid-latitudes. This is because the wind contribution increased in this region, which led to a weakening relative contribution of internal tides. Compared with the internal tide conversion rates, the pattern of $K_z$ was inconsistent with internal tides, even at lower latitudes. It can be inferred that the turbulent mixing was not only affected by the internal tides, but also by other factors. There is a strong western boundary flow, Kuroshio, and an active mesoscale environment in this region. Some researchers have shown that the existence of mesoscale environment will alter the internal tide features, so we reasonably infer that the tidal induced turbulent mixing in this area was modulated by the mesoscale features.

[Figure]

**Figure 7 Scatter of log-scale $K_z$ versus log-scale internal tide conversion rate in 250-500 m (row 1), 500-1000**

**m (row 2), 1000-15000 m (row 3) and the best-fit slopes are denoted by the red line. Columns 1 and ,2,3 are**

**10°N-125°N, 15°N 25°N and 25°N -35°N latitude bands, the 95% confidence interval is indicated by dash**

**lines.respectively.**

**3.4 Role of Mesoscale features in tidal mixing**

Focusing on the low latitudes, where tidal mixing is dominated, the diapycnal diffusivities, $K_z$, related to internal tides and eddy kinetic energy are shown in (Fig.8). The combined influences of mesoscale features and internal tides on mixing are indicated. The increasing internal tide conversion rates significantly enhanced turbulent mixing. We find a correlation between elevated eddy kinetic energy and the averaged diapycnal diffusivities for a given internal tide conversion rate level.

[revised manuscript text omitted]

Zhao,Z., M. H. Alford, J. A. MacKinnon, and R. Pinkel: Long-range propagation of the semidiurnal
internal tide from the Hawaiian Ridge. J. Phys. Oceanogr., 40, 713–736, 2010.

Zhao, Z.: Mapping internal tides from satellite altimetry without blind directions. Journal of
Geophysical Research: Oceans, 124, 2019. https://doi.org/10.1029/2019JC015507

---

## Author Comment (AC2)

**Referee #2**

The present manuscript describes spatial pattern and seasonal variability of the diapycnal diffusivities in the Philippine Sea. It was shown that seasonal variability was strong in winter and weak in summer at mid-latitudes, with the seasonal fluctuations more obvious in the upper ocean. The diapycnal diffusivitie that is spatially inhomogeneous were estimated from ARGO float data with the fine scale parameterization. The present manuscript is good scientific quality and well written.

*First of all, thank you for your support to our work, we have carefully considered your advises and revised the manuscript.*

The obtained results are interesting however revision is needed:

1. More convincing comparison and analysis is needed for diapycnal diffusivities scatters fig 6-7.

*Response:*

*We have added some detailed descriptions and marked 95% confidence interval in these figures (see section 3.3.2). Figs. 6 and 7 are used to support the conclusions from Fig. 4 and Fig. 5. The regression coefficient can represent the response of mixing to wind (eg. Qiu et al., 2012) or other factors(eg. Wu et al., 2011; Jeon et al., 2018). And the same conclusion can be drawn from the two kinds of analysis methods.*

2. As far as in Fig.3 (diapycnal diffusivities) and Fig.5 (Vertical structures of geometric averaged diapycnal diffusivities) Philippine Sea was divided for two zones (a) 10°N -25°N and (b) 25°N-35°N, but on figures 6-7 Philippine Sea was divided into three zones 10°N -15°N, 15°N-25°N and 25°N-35°N it is difficult to compare the results for zone (10-25) and make a conclusions about that results on Figs. 6-7 is consistent with the results of Fig.3 and Fig.4.

*Response:*

*Good suggestion. According to your comment, we reprocessed the data and redrew the figures. We divided the region into two parts, low latitude and mid-latitude, which are consistent with the division in Fig. 4 and Fig. 5. We found that the new division does not affect the conclusion but can actually interpret the results better. The new Fig. 6 and Fig.7 have been added in main text and corresponding contexts were revised.*

[Figure]

**Figure 1 Scatter of log-scale Kz versus log-scale near-inertial energy flux from wind in 250-500 m between (a) 10°N -25°N and (b) 25°N-35°N, and in 500-1000 m between (c) 10°N-25°N and (d) 25°N-35°N The best-fit slopes are denoted by the solid line, the 95% confidence interval is indicated by dash lines.**

[Figure]

**Figure 2 Scatter of log-scale $K_z$ versus log-scale internal tide conversion rate in 250-500 m (row 1), 500-1000**

**m (row 2), 1000-15000 m (row 3) and the best-fit slopes are denoted by the red line. Columns 1 and    are 10°N-25°N and 25°N -35°N latitude bands, the 95% confidence interval is indicated by dash lines.**

3. In line 182 H is described as is the mixed-layer depth and was set to a constant 25m, however in Eq (8) H – near-inertial energy flux.

*Response:*

*Thank you for your attention, it has been revised.*

4. Typo in Figure 3 Seasonal cycles in diapycnal diffusivities (colorful line) and near-inertial energy flux from wind (green) extents to 250-500 m, 500-1000 m and 1000-1500 m in (a) 10°N -25°N and (b) 10°N-25°N (should be 25°N -35°N ).

*Response:*

*It has been corrected.*

*References:*

Wu, L., Jing, Z., Riser, S., & Visbeck, M.: Seasonal and spatial variations of southern ocean diapycnal mixing from argo profiling floats, Nature Geoscience, 2011.

Jeon, C., Park, J.H., & Park, Y.G.: Temporal and spatial variability of near‐inertial waves in the East/Japan Sea from a high‐resolution wind‐forced ocean model, 
[revised manuscript text omitted]

Each 1°latitude band was separated into two regions with weak or strong internal tides (or other factors). Here we define the strong or weak internal tides as the internal tide conversion rate larger or smaller than the median of the Philippine Sea. The diapycnal diffusivities in these two kinds of regions were then averaged. The similar method has been used to analyze the effect of topography and different frequency-bands internal waves on ocean interior shear and mixing (eg. Whalen et al., 2012; Zhang et al., 2019). For a more direct representation, the ratios of diapycnal diffusivities above the strong internal tides to weak internal tides were shown. The ratio larger than 1 means that the diapycnal diffusivities are significantly higher in the regions of strong internal tides.

The larger this ratio is ,the more important the internal tidal induced mixing. Similarly, the contributions of near-inertial wave and eddy kinetic energy were analyzed by this statistical method. The strong near-inertial wave or eddy kinetic energy is the locations where this parameter exceeds the regional median.

[Figure]

**Figure 4 Ratios of diapycnal diffusivities between areas over strong (greater than median) and weak internal tide (red lines), strong (greater than median) and weak near-inertial wave (green lines) and strong (greater than median) and weak eddy kinetic energy (blue lines) for each 1° latitude bands in the depth range of (a) 250-500 m, (b) 500-1000 m and (c) 1000-1500 m, Which averages for each bands containing more than 10 estimates.**

At depths of 250-500 m, the  ratio associated with internal tide increased significantly at 10°N,

21°N and 33°N. These latitudes correspond to Guap seamount, Luzon Strait and Izu Ridge, which are main internal tide source sites. It reached 2 near these three latitudes, indicating that strong internal tides triggered the enhancement of $K_z$ twice as much compared to the regions of weak internal tides. In addition, north of 23°N, the ratio in related to NIW in the upper ocean increased significantly with latitude,  which indicated that the wind plays a more important role in mixing at this latitude band. This result is basically consistent with previous studies (Whalen et al., 2018), which suggested that the mixing is dominated by wind  in the mid-latitude. Taking the wind as the driving factor better explains the seasonal cycle of diapycnal diffusivities in Fig.3, since the winds have an apparent seasonal dependence. The obvious seasonal trend of $K_z$ due to the important contribution of wind occurs between 25°N-35°N. In contrast, the ratio for wind is ~1 at lower latitudes, indicating that the wind-driven mixing is insignificant here with the absence of wind-driven seasonal cycle.

The  wind contribution to turbulent mixing is significantly reduced in the depth ranges of 500-1000 m and 1000-1500 m (Fig.4 b and c). The ratio only increased slightly at mid-latitudes, less than 2 anywhere. In contrast, the enhancement of mixing triggered by internal tides at these depth ranges was more significant, with the ratios exceeding 3.5 at some latitudes. This suggested that internal tides played a more important role in deep ocean mixing. Furthermore, internal tides significantly enhanced $K_z$ around 13°N, 21°N, and 29°N, corresponding to the sources of Mariana Trench, Luzon

Strait and Bonin Ridge, respectively. Such enhancement was not obvious at the Izu Ridge possibly due to the shallower depth and paucity of deep data, or the turning latitude effects in this area.

Combined with the analysis of relative contributions of different factors in different layers, it was concluded that the contribution of internal tides in turbulent mixing is more important in low latitudes of the Philippine Sea. In this area, the wind and mesoscale features did not significantly enhance $K_z$. At mid-latitudes, internal tides still played an important role, but the wind contribution was more significant in the upper ocean. The wind drove turbulent mixing even at the depths of 500-1000 m and 1000-1500

m. The mid-latitude region not only corresponds to westerlies, but also features energetic mesoscale motions. Therefore, the mesoscale features might be a potential factor for the enhanced turbulent mixing.

The modulation of mesoscale environments in the wind-induced mixing are discussed by some previous studies (eg. Jing et al., 2011; Whalen et al., 2018), while the impact of mesoscale features in tide-induced mixing and in lower latitudes were not be considered.

The Philippine Sea was separated into two latitude bands. The vertical structures of diapycnal diffusivities in the regions with strong or weak internal tides were compared (Fig.5). This result can directly reveal the enhancement of internal tide on mixing at different depths. A similar analysis was used for wind and EKE. In the low latitudes, $K_z$ did not increase in the regions of high eddy kinetic energy or strong near-inertial energy, whereas, it increased significantly in the regions of strong internal tides. This enhancement was more obvious below 400 m (Fig.5 a). And in the mid-latitudes, $K_z$

in the upper ocean increased significantly corresponding to strong winds with compared to weak winds (Fig.5 b). Meanwhile, $K_z$ was also larger in the regions of strong internal tides and high EKE in the upper ocean. The enhancement of wind or EKE to turbulent mixing significantly weakened below 600

m, while the enhancement of internal tides increased with depth. Here, these results convey several information: 1. Wind and EKE play  important roles on mixing in the upper ocean in the middle latitudes; 2. Strong internal tide facilitate and enhance mixing in the deeper ocean. These two conclusions are consistent with previous researchers (eg. Jing et al., 2011; Whalen et al., 2012;

Waterhouse et al., 2014; Mackinnon et al., 2017; Whalen et al., 2018). In addition, our results indicate that, in the Philippine Sea, internal tides play a significant role in turbulent mixing not only in the low latitudes, but also in the mid latitudes,and not only in the deeper ocean, but also in the upper ocean.

[Figure]

**Figure 5 Vertical structures of geometric averaged diapycnal diffusivities $K_z$ with weak and strong wind**

**(green), low and high EKE (blue) and weak and strong internal tide (red) in the (a) low-latitude and (b)**

**middle latitudes.**

**3.3.2 Wind**

We adopted the linear regression approach and obtained the correlation between diapycnal diffusivities and wind. This approach is generally used to statistics the correlation between two factors (eg. Wu et al.,

2011; Jing et al., 2014; Jeon et al., 2018; Zhao et al., 2019). The least squares linear fit slopesregression coefficient is able to of $K_z$ to NIW energy from wind represent the mixing response to wind (eg. Qiu et al., 2012). Here, the Philippine Sea is divided into 10°N 15°N, 15°N-25°N and 25°N-35°N (Fig.6). At the depth of 250-500 m, the slope is the largessignificant largeert in 25°N-35°N (~0.305), followed by that in 10°N 15°N (~0.133), and the smalleerst in 150°N-25°N (~0.01329). The wind driven turbulent mixing was most significant between 25 °N-35 °N, but was insignificant between 150°N-25°N. At the depth of 500-1000 m, the wind influence on turbulent mixing was weakened in the mid-latitudes. This was consistent with the results of Fig.3 and Fig.4. It proved that the contribution of wind has a latitudinalzonal dependence, which was significant at the mid-latitudes, but insignificant at low latitudes.

In addition, the response of turbulent mixing to wind weakened quickly with depth, indicating that the dominant factor of mixing in the deeper water column was not wind. Accordingly, it was difficult for wind to drive mixing below 1000 m, so we do not show the results at the depth of 1000-1500 m (Fig.4).

[Figure]

**Figure 6 Scatter of log-scale Kz versus log-scale near-inertial energy flux from wind in 250-500 m between (a)**

**10°N -2 , (b) 15°N-25°N and (e) 25°N-35°N, and in 500-1000 m between (d) 10°N-2 , (e) 15°N-25°N**

**and (f) 25°N-35°N The best-fit slopes are denoted by the solid line, the 95% confidence interval is indicated**

**by dash lines.**

**3.3.3 Tide**

The slopes of $K_z$ to internal tidal conversion rates represent the mixing response to internal tides. As discussed above, the mixing significantly responded to the internal tides over the entire Philippine Sea (Fig.7). The relationship was depth dependent. The slopes did not reach 0.1 at the depth of 250-500 m, but increased significantly at 500-1000 m and 1000-1500 m. and reach 0.128 for the deepest depth band. The response of mixing to internal tides was more significant in the deeper ocean. Focusing on different latitude bands, the slopes of $K_z$ to internal tides is smaller at mid-latitudes. This is because the wind contribution increased in this region, which led to a weakening relative contribution of internal tides. Compared with the internal tide conversion rates, the pattern of $K_z$ was inconsistent with internal tides, even at lower latitudes. It can be inferred that the turbulent mixing was not only affected by the internal tides, but also by other factors. There is a strong western boundary flow, Kuroshio, and an active mesoscale environment in this region. Some researchers have shown that the existence of mesoscale environment will alter the internal tide features, so we reasonably infer that the tidal induced turbulent mixing in this area was modulated by the mesoscale features.

[Figure]

**Figure 7 Scatter of log-scale $K_z$ versus log-scale internal tide conversion rate in 250-500 m (row 1), 500-1000**

**m (row 2), 1000-15000 m (row 3) and the best-fit slopes are denoted by the red line. Columns 1 and ,2,3 are**

**10°N-125°N, 15°N 25°N and 25°N -35°N latitude bands, the 95% confidence interval is indicated by dash**

**lines.respectively.**

**3.4 Role of Mesoscale features in tidal mixing**

Focusing on the low latitudes, where tidal mixing is dominated, the diapycnal diffusivities, $K_z$, related to internal tides and eddy kinetic energy are shown in (Fig.8). The combined influences of mesoscale features and internal tides on mixing are indicated. The increasing internal tide conversion rates significantly enhanced turbulent mixing. We find a correlation between elevated eddy kinetic energy and the averaged diapycnal diffusivities for a given internal tide conversion rate level.

[revised manuscript text omitted]

Wu, L. , Jing, Z. , Riser, S. , & Visbeck, M.: Seasonal and spatial variations of southern ocean diapycnal mixing from argo profiling floats. Nature Geoscience, 2011.

Wunsch, C., and R. Ferrari: Vertical mixing, energy and the general circulation of the oceans. Annu. Rev.

Fluid Mech., 36, 281–314, 2004.

Xu, Z., Liu, K., Yin, B., Zhao, Z., Wang, Y., & Li, Q.: Long-range propagation and associated variability of internal tides in the South China Sea, Journal of Geophysical Research: Oceans, 121,

8268–8286, 2016.

Xu, Z., Yin, B., Hou, Y., & Liu, A. K.: Seasonal variability and north–south asymmetry of internal tides in the deep basin west of the Luzon Strait, Journal of Marine Systems, 134, 101–112, 2014.

Xu, Z., Yin, B., Hou, Y., & Xu, Y.: Variability of internal tides and near-inertial waves on the continental slope of the northwestern South China Sea, Journal of Geophysical Research: Oceans,

118, 197–211, 2013.

Zhang, Z., Qiu, B., Tian, J. et al.: Latitude-dependent finescale turbulent shear generations in the Pacific tropical-extratropical upper ocean.Nat Commun 9, 4086, 2018.

Zhao,Z., M. H. Alford, J. A. MacKinnon, and R. Pinkel: Long-range propagation of the semidiurnal internal tide from the Hawaiian Ridge. J. Phys. Oceanogr., 40, 713–736, 2010.

Zhao, Z.: Mapping internal tides from satellite altimetry without blind directions. Journal of

Geophysical Research: Oceans, 124, 2019. https://doi.org/10.1029/2019JC015507

---

## Author Response (AR1)

**Referee #1**

The paper deals with an interesting and important problem, namely the mechanism(s) responsible for oceanic mixing in a topographically-complex region. The data have been processed satisfactorily and they show some interesting features, though the interpretation of these features is not straightforward. My principal difficulty with the paper lies in the analysis of the data and the validity of the conclusions drawn from such an analysis. The heart of the analysis is the correlation of observed features of the diapycnal diffusivity patterns with the main causes of mixing identified by the authors. Many of the correlations claimed by the authors seem to be based either on visual inspection of the data plots or analysis based upon ratios of the energy fluxes associated with each of the driving agencies (e.g Fig 4) . I did not find these analyses convincing and I would have appreciated some more quantitative and rigorous data correlation procedures carried out to justify the conclusions. Figs 6, 7 and 8 are very difficult to interpret. Overall, I found the data analysis to be rather superficial and, in consequence, the conclusions unjustified by the evidence of the data analysis.

*Response:*

*Thank you for your time and constructive comments. The convincing verifications of both the methods and results are crucial for a scientific paper. According to your helpful suggestions, we have thoroughly made the validation of the analysis methods used in present manuscript. The detailed descriptions and discussions were added to section 3.3.1 (Fig. 4 and Fig. 5), section 3.3.2 (Fig. 6 and Fig. 7) and section 3.4 (Fig. 8). Based on the verified methods and the comparison to other studies, the conclusions that have been drawn are found to be reliable and convincing.*

*1. As for the parameterization method itself, it is widely used in the studies of ocean mixing. The most beneficial point is that there are amount of observations of temperature and salinity by ARGO among the global ocean, with good spatial and temporal coverage. The parameterization method was used to estimate global or regional distributions of diapycnal mixing, which can supplement the scarcity of direct turbulence observations. The parameterization analysis seems not straightforward but effective in mixing investigation through several statistical analysis and correlation analysis with other impact factors (e.g. Wu et al., 2011; Whalen et al., 2012; Kunze et al., 2017; Chanona et al., 2018).*

*2. As for the statistical and correlation analysis methods*

*a. To address the relative contributions to the mixing from possible factors, we showed Fig. 4 and Fig. 5. Each 1° latitude band was separated into two regions with weak or strong internal tides (or other factors). Here we define the strong or weak internal tides as the internal tide conversion rate larger or smaller than the median internal tide conversion rate of the Philippine Sea (for the details see section 3.3.1). The diapycnal diffusivities in these two kinds of regions were then averaged. The similar method has been used to analyze the effect of topography and different frequency-bands internal waves on ocean interior shear and mixing (eg. Whalen et al., 2012; Zhang et al., 2018). For a more direct representation, the ratios of diapycnal diffusivities above the strong internal tides to weak internal tides were shown. The ratio larger than 1 means that the diapycnal diffusivities are significantly higher in the regions of strong internal tides. The similar analysis were used for the analysis of wind and EKE effects.*

[Figure]

**Figure 4 Ratios of diapycnal diffusivities between areas over strong (greater than median) and weak internal tide (red lines), strong (greater than median) and weak near-inertial wave (green lines) and strong (greater than median) and weak eddy kinetic energy (blue lines) for each 1° latitude bands in the depth range of (a) 250-500 m, (b) 500-1000 m and (c) 1000-1500 m, Which averages for each bands containing more than 10 estimates.**

[Figure]

(Whalen et al., GRL, 2012, Figure 3a ) E.g. Global mean dissipation rate for 3_x0005_ half-overlapping latitudinal bands in the depth range 250–1,000 m over rough (variance greater than global mean) and smooth topography with 90% bootstrapped confidence intervals.

[Figure]

Figure 5 Vertical structures of geometric averaged diapycnal diffusivities $K_z$ with weak and strong wind (green), low and high EKE (blue) and weak and strong internal tide (red) in the (a) low-latitude and (b) middle latitudes.

[Figure]

(Chanona et al., JGR, 2018, Figure 11 ) E.g. (a) Spatial distribution of strong (purple) and weak (blue) barotropic tidal speeds (b) spatial distribution of rough (purple) and smooth (blue) topography as determined by topographic roughness, and (c) log-scale histogram of stormy (purple) and calm (blue) 10 m wind speeds, associated with each conductivity-temperature-depth profile. High/low bin cutoffs for each parameter are defined by the upper/lower quartile limits of their corresponding distributions; gray regions indicate interquartile values not included in our analysis. Average vertical profifiles of $\varepsilon_{IW}$ (middle column) and $K_{IW}$ (right column) are binned accordingly for (d, g) strong/weak tidal speeds, (e, h) rough/smooth topography, and (f, i) stormy/calm wind speeds. The number of profiles comprising each vertical average is given by n and shading indicates the associated geometric standard deviations.

*b. In order to further verify our results of Fig. 4 and. 5, the Fig. 6 and Fig. 7 were plotted using another direct correlation analysis method. The same conclusion can be drawn from the two kinds of analysis methods. The linear regression is a popular method for statistics analysis of the correlation between two factors (eg. Wu et al., 2011; Jing et al., 2014; Jeon et al., 2018; Zhao et al., 2019). The regression coefficient can represent the response of mixing to wind (eg. Qiu et al., 2012) or other factors. Another reviewer also gave some advises for comparing and interpreting these figures. Noted that Fig. 6 and Fig. 7 have been revised according to the suggestions from both reviewers.*

[Figure]

**Figure 6 Scatter of log-scale Kz versus log-scale near-inertial energy flux from wind in 250-500 m between (a) 10°N -25°N and (b) 25°N-35°N, and in 500-1000 m between (c) 10°N-25°N and (d) 25°N-35°N The best-fit slopes are denoted by the solid line, the 95% confidence interval is indicated by dash lines.**

[Figure]

**Figure 7** Scatter of log-scale $K_z$ versus log-scale internal tide conversion rate in 250-500 m (row 1), 500-1000 m (row 2), 1000-15000 m (row 3) and the best-fit slopes are denoted by the red line. Columns 1 and are 10°N-25°N and 25°N -35°N latitude bands, the 95% confidence interval is indicated by dash lines.

*3. As for the conclusions*

*The main conclusions in this paper include: 1. internal tides played a significant role in mixing the whole water column in the Philippine Sea. 2. the energetic mesoscale environment increases the internal tidal mixing.*

*We believe our conclusions are reliable and convincing, with several reasons:*

*a. The methods used are reasonable. The statistical methods have all passed the significance test.*

*b. We used different methods for the analysis and come to the same conclusion (eg. Figs. 4,5 and Figs. 6,7; Fig. 8 and Fig. 9). The results from different methods can support each other.*

*c. Some conclusions can be supported by previous studies. For example, according to Fig. 4 and Fig. 5, we can get the following three messages: 1. Wind and EKE play important roles on mixing the upper ocean in the mid-latitudes; 2. Strong internal tides are easier to enhance mixing in the deeper ocean. 3. Internal tides played a significant role in turbulent mixing not only in the low latitudes, but also in the mid latitudes with r strong winds. The points 1 and 2 are consistent with some previous works (e.g. Waterhouse et al., 2014; Mackinnon et al., 2017; Whalen et al., 2018). The point 3 can be explained by some theoretical and numerical studies. For example, the eddy flows can increase vertical shear and subsequently internal tide energy dissipation rate (eg. Chavanne et al., 2010, Dunphy, 2014).*

*Therefore, we carefully considered your comments, checked and verified our methods and results again. We believe our methods used are valid and our main conclusions are reliable and reasonable.*

There are a few typographic and/or grammatical errors but, in general, the standard of English is satisfactory. The main error is to use "rates" instead of "ratios" in the

correlation analyses (line 250 and elsewhere). The use of "slopes" instead of "ratios" (e.g line 319 and elsewhere) is misleading also (if I have understood the text correctly). In equation (3) I assume that "acrcosh" should be "arccosh"?

**Response:**

*Thank you for your comments, the "rates" (line 250 and elsewhere) was misused in this paper, and it has been corrected. The Figs. 7 and 8 used the linear regression approach, so we think the "slopes" is more suitable here. And the error in equation (3) has been corrected.*

**References:**

Chanona, M., Waterman, S. and Gratton, Y.: Variability of internal wave-driven mixing and stratification in Canadian Arctic shelf and shelf-slope waters, Journal of Geophysical Research: Oceans, 123, 9178–9195, 2018.

Chavanne, C., Flament, P., Luther, D., & Gurgel, K. W.: The surface expression of semidiurnal internal tides near a strong source at Hawaii. Part Ⅱ: interactions with mesoscale currents*, Journal of

Physical Oceanography, 40(6), 1180-1200, 2010.

Dunphy, M., and K. G. Lamb: Focusing and vertical mode scattering of the first mode internal tide by mesoscale eddy interaction, J. Geophys. Res. Oceans, 119, 523–536, 2014.

Jing, Z., Wu, L., Li, L., Liu, C , Liang, X., & Chen, Z., et al.: Turbulent diapycnal mixing in the subtropical northwestern pacific: spatial-seasonal variations and role of eddies, Journal of Geophysical Research Oceans, 116, 2011.

Jeon, C., Park, J.H., & Park, Y.G.: Temporal and spatial variability of near‐inertial waves in the East/Japan Sea from a high‐resolution wind‐forced ocean model, Journal of Geophysical Research: Oceans, 124, 6015–6029, 2018. https://doi.org/10.1029/2018JC014802

Kunze, E.: Internal-wave-driven mixing: global geography and budgets, Journal of Physical Oceanography, JPO-D-16-0141.1, 2017.

MacKinnon, J. A. et al.: Climate process team on internal-wave driven ocean mixing, Bull. Am. Meteorol. Soc. 98, 2429–2454, 2017.

Qiu, B., Chen, S., and Carter, G. S.: Time‐varying parametric subharmonic instability from repeat CTD surveys in the northwestern Pacific Ocean, J. Geophys. Res., 117, C09012, 2012.

Waterhouse, A.F., J.A. MacKinnon, J.D. Nash, M.H. Alford, E. Kunze, H.L. Simmons, K.L. Polzin, L.C. St. Laurent, O.M. Sun, R. Pinkel, L.D. Talley, C.B. Whalen, T.N. Huussen, G.S. Carter, I. Fer, S. Waterman, A.C. Naveira Garabato, T.B. Sanford, and C.M. Lee: Global Patterns of Diapycnal Mixing from Measurements of the Turbulent Dissipation Rate. J. Phys. Oceanogr., 44, 1854–1872, 2014

Wu, L., Jing, Z., Riser, S., & Visbeck, M.: Seasonal and spatial variations of southern ocean diapycnal mixing from argo profiling floats, Nature Geoscience, 2011.

Whalen, C. B., Talley, L. D., & Mackinnon, J. A.: Spatial and temporal variability of global ocean mixing inferred from ARGO profiles. Geophysical Research Letters, 39(18), 2012.

Whalen, C.B., MacKinnon, J.A. & Talley, L.D.: Large-scale impacts of the mesoscale environment on mixing from wind-driven internal waves. Nature Geosci 11, 842–847, 2018.

Zhang, Z., Qiu, B., Tian, J. et al.: Latitude-dependent finescale turbulent shear generations in the Pacific tropical-extratropical upper ocean, Nat Commun 9, 4086, 2018.

Zhao, Z.: Mapping internal tides from satellite altimetry without blind directions, Journal of Geophysical Research: Oceans, 124, 2019.

**Referee #2**

The present manuscript describes spatial pattern and seasonal variability of the diapycnal diffusivities in the Philippine Sea. It was shown that seasonal variability was strong in winter and weak in summer at mid-latitudes, with the seasonal fluctuations more obvious in the upper ocean. The diapycnal diffusivitie that is spatially inhomogeneous were estimated from ARGO float data with the fine scale parameterization. The present manuscript is good scientific quality and well written.

*First of all, thank you for your support to our work, we have carefully considered your advises and revised the manuscript.*

The obtained results are interesting however revision is needed:

1. More convincing comparison and analysis is needed for diapycnal diffusivities scatters fig 6-7.

*Response:*

*We have added some detailed descriptions and marked 95% confidence interval in these figures (see section 3.3.2). Figs. 6 and 7 are used to support the conclusions from Fig. 4 and Fig. 5. The regression coefficient can represent the response of mixing to wind (eg. Qiu et al., 2012) or other factors(eg. Wu et al., 2011; Jeon et al., 2018). And the same conclusion can be drawn from the two kinds of analysis methods.*

2. As far as in Fig.3 (diapycnal diffusivities) and Fig.5 (Vertical structures of geometric averaged diapycnal diffusivities) Philippine Sea was divided for two zones (a) 10°N -25°N and (b) 25°N-35°N, but on figures 6-7 Philippine Sea was divided into three zones 10°N -15°N, 15°N-25°N and 25°N-35°N it is difficult to compare the results for zone (10-25) and make a conclusions about that results on Figs. 6-7 is consistent with the results of Fig.3 and Fig.4.

*Response:*

*Good suggestion. According to your comment, we reprocessed the data and redrew the figures. We divided the region into two parts, low latitude and mid-latitude, which are consistent with the division in Fig. 4 and Fig. 5. We found that the new division does not affect the conclusion but can actually interpret the results better. The new Fig. 6 and Fig.7 have been added in main text and corresponding contexts were revised.*

[Figure]

**Figure 1 Scatter of log-scale Kz versus log-scale near-inertial energy flux from wind in 250-500 m between (a) 10°N -25°N and (b) 25°N-35°N, and in 500-1000 m between (c) 10°N-25°N and (d) 25°N-35°N The best-fit slopes are denoted by the solid line, the 95% confidence interval is indicated by dash lines.**

[Figure]

**Figure 2 Scatter of log-scale $K_z$ versus log-scale internal tide conversion rate in 250-500 m (row 1), 500-1000**

**m (row 2), 1000-15000 m (row 3) and the best-fit slopes are denoted by the red line. Columns 1 and are 10°N-25°N and 25°N -35°N latitude bands, the 95% confidence interval is indicated by dash lines.**

3. In line 182 H is described as is the mixed-layer depth and was set to a constant 25m, however in Eq (8) H – near-inertial energy flux.

*Response:*

*Thank you for your attention, it has been revised.*

4. Typo in Figure 3 Seasonal cycles in diapycnal diffusivities (colorful line) and near-inertial energy flux from wind (green) extents to 250-500 m, 500-1000 m and 1000-1500 m in (a) 10°N -25°N and (b) 10°N-25°N (should be 25°N -35°N ).

*Response:*

*It has been corrected.*

*References:*

Wu, L., Jing, Z., Riser, S., & Visbeck, M.: Seasonal and spatial variations of southern ocean diapycnal mixing from argo profiling floats, Nature Geoscience, 2011.

Jeon, C., Park, J.H., & Park, Y.G.: Temporal and spatial variability of near‐inertial waves in the East/Japan Sea from a high‐resolution wind‐forced ocean model, Journal of Geophysical Research: Oceans, 124, 6015–6029, 2018. https://doi.org/10.1029/2018JC014802

Qiu, B., Chen, S., and Carter, G. S.: Time‐varying parametric subharmonic instability from repeat CTD surveys in the northwestern Pacific Ocean, J. Geophys. Res., 117, C09012, 2012.